# RETHINKING PERTURBATION PREDICTION BASELINES

**Junwei Sun**
Department of Biostatistics
Johns Hopkins University
Baltimore, MD 21205, USA
jsun140@jh.edu

**Ouyang Zhu**
Department of ACMS
University of Notre Dame
Notre Dame, IN 46556, USA
ozhu@nd.edu

**Yiqun T. Chen**
Departments of Biostatistics and Computer Science
Johns Hopkins University
Baltimore, MD 21205, USA
yiqunc@jhu.edu

## ABSTRACT

Predicting cellular responses to genetic perturbations is central to understanding biology and unlocking more efficient drug and genetic therapy discovery. Recent approaches leverage large language models and deep learning for this task, yet simple baselines for predicting categorical outcomes — such as whether a gene is differentially expressed or up- or down-regulated — remain underexplored. We evaluate two simple baselines on Perturb-seq screens from four cell lines: a gene-based majority vote and an embedding-based $k$-nearest neighbors classifier. On curated benchmarks, majority vote alone achieves accuracies of 0.62–0.80, but collapses on full, unfiltered data, exposing how dataset curation can inflate model performance. On the same unfiltered data, a nearest-neighbor classifier matches LLM-based methods and remains competitive with state-of-the-art deep generative models in cross-cell-line transfer tasks. These results highlight the need for stronger baselines and for directly modeling categorical perturbation outcomes.

## 1 INTRODUCTION

Predicting how cells respond to drug and genetic perturbations is a key step toward building "virtual cells" that could accelerate drug discovery and precision medicine (Rood et al., 2024; Bunne et al., 2024). This goal is supported by recent technologies such as Perturb-seq (Dixit et al., 2016), which combines CRISPR-based perturbations with single-cell RNA-seq to provide interventional data on the effects of gene knockdowns on gene expression (e.g., whether a gene is differentially expressed (DE), and if so, whether it is up- or down-regulated). For instance, perturbing *ANK2*, a gene linked to neurodevelopment, led to significantly increased expression of an interneuron-specific gene module, suggesting a potential role in interneuron maturation beyond its known functions (Jin et al., 2020).

With the rise of generative AI, recent work has turned to building genomic models using pretraining strategies from large language models (Cui et al., 2023; Roohani et al., 2023), as well as directly querying LLMs such as GPT-4o/o1 and LLaMA (Wu et al., 2025; Phillips et al., 2025; Swanson et al., 2025). Alongside these modeling efforts, the importance of *careful evaluation* has become apparent. One line of work has proposed *simple yet effective baselines*, such as regularized regression, to predict the *average effect* of perturbations, showing that deep learning models often fail to outperform linear baselines on standard metrics like mean squared error (MSE) and correlation (Ahlmann-Eltze et al., 2025). Others have scrutinized evaluation practices more directly, calling for better train/test splits and distributional metrics beyond mean accuracy (Viñas Torné et al., 2025). However, most of this work has focused on predicting **continuous gene expression values**. Comparatively little attention has been given to **predicting categorical outcomes – whether a gene is differentially expressed, up-regulated, or down-regulated** – which are often easier to interpret biologically.

In this work, we seek to probe beyond continuous prediction metrics, as absolute transcript abundance is driven by technical rather than biological artifacts – such as platform, protocol, and normalization. In particular, we ask: (1) How well do current models perform when we convert predicted continuous gene expression values into qualitative differential expression categories? and (2) How well do models perform when we directly predict categorical DE outcomes?

Closest to our work are PerturbQA (Wu et al., 2025) and SynthPert (Phillips et al., 2025), both of which leverage LLMs to integrate prior information through retrieval-augmented generation and reasoning traces, respectively. We explore simpler approaches: direct prediction using embedding vectors in $k$-nearest neighbor or gene-based majority vote models. **On benchmark datasets, these methods outperform many LLM-based approaches and continuous prediction models.** Based on our findings, we call for greater attention to downstream objectives of virtual cell models. If the goal is to predict differential expression status or changes in cell states, it may be more interpretable to model these categorical outcomes directly. Building more expressive models on our simple baselines may also yield more performant methods.

## 2 METHODS

Following Phillips et al. (2025), we formulate perturbation outcome prediction as a three-class classification problem. Given a perturbation–gene pair $(P, G)$, the task is to predict whether gene $G$ under perturbation $P$ is: (1) non-differentially expressed (`Non-DE`), (2) up-regulated (`Up`), or (3) down-regulated (`Down`). An alternative approach is to treat these as two sequential binary classification problems (Wu et al., 2025), and we found qualitatively similar conclusions. Code used to generate the analysis in this paper can be found at `https://github.com/ivysun14/Project-Perturb`.

### 2.1 DATASET, PROCESSING, AND SAMPLING

We benchmark on essential gene screens from K562, RPE1, Jurkat, and HepG2 cell lines, chosen for their widespread use in perturbation effect prediction (Nadig et al., 2025; Replogle et al., 2022), which facilitates direct comparison across methods. For each dataset, we followed prior work (Roohani et al., 2023; Adduri et al., 2025) and normalized raw UMI counts by dividing by the total counts per cell, scaling to 10,000, and applying a $\log(1 + x)$ transformation. To obtain DE statistics, we performed a Wilcoxon rank-sum test with Benjamini–Hochberg (BH) adjusted p-value per perturbation, controlling the false discovery rate at $p < 0.01$. For DE pairs, the sign of the log-fold change determines up- or down-regulation (see Table 1).

We consider two evaluation settings based on how datasets are curated. First, we evaluate models on **PerturbQA-downsampled datasets** where in Wu et al. (2025) non-DE pairs were downsampled to address the severe class imbalance present across all four datasets (95–98% non-DE), which can bias both training and evaluation, and DE pairs were restricted to those with top statistical significance. Second, we evaluate on the **full dataset without downsampling**, reflecting the more realistic setting where no prior knowledge of class distribution is available. We annotate our results to distinguish between these two settings. In both cases, we use a 75/25 train/test split such that all test perturbations are unseen during training.

### 2.2 WITHIN CELL LINE MODELING

**Perturbation Majority Vote.** We first consider a simple majority vote rule based on the hypothesis that a gene's response to perturbation largely reflects its intrinsic properties rather than the specific perturbation applied. That is, if gene $G$ tends to be up-regulated across most training perturbations, we predict it will also be up-regulated under a new, unseen perturbation. Formally, for each gene $g$, we assign the most frequent response class observed across all training perturbations:

$$\hat{y}_g = \text{mode}\left(\{y_{g,p}\}_{p \in \mathcal{P}_{\text{train}}}\right), \quad y_{g,p} \in \{\text{Up}, \text{Down}, \text{Non-DE}\}, \tag{1}$$

where $\mathcal{P}_{\text{train}}$ denotes the set of training perturbations and $y_{g,p}$ is the observed DE status of gene $g$ under perturbation $p$. Notably, because this baseline ignores perturbation identity entirely, it produces the same prediction for gene $g$ regardless of which perturbation is applied. Models that fail to outperform this baseline may not be capturing meaningful perturbation-specific signal.

Table 1: Perturbation dataset statistics.

|  | K562 | RPE1 | HepG2 | Jurkat |
|---|---|---|---|---|
| Total pairs (M) | 5.5 | 7.7 | 23.0 | 21.3 |
| Non-DE (%) | 91.1 | 89.6 | 96.7 | 97.3 |
| Up-reg (%) | 3.86 | 4.72 | 1.41 | 0.82 |
| Down-reg (%) | 5.00 | 5.69 | 1.91 | 1.90 |
| Train pairs (M) | 3.3 | 4.6 | 13.8 | 12.7 |
| Test pairs (M) | 2.2 | 3.1 | 9.2 | 8.5 |
| Perturbations | 1,092 | 1,543 | 2,393 | 2,393 |
| Genes | 4,999 | 5,000 | 9,623 | 8,881 |

**Weighted $k$-Nearest Neighbors.** To incorporate perturbation-specific information, we use a weighted $k$-nearest neighbors (kNN) classifier with GenePT embeddings (Chen & Zou, 2023) to represent perturbations. For each query pair $(P, G)$, we identify the $k$ most similar training perturbations via Euclidean distance in embedding space (or equivalently the cosine similarity, since the embeddings are $\ell_2$-normalized) and predict $G$'s response through distance-weighted voting. This baseline evaluates whether local neighborhoods in perturbation embedding space capture sufficient information for predicting gene responses.

## 2.3 CROSS-CELL-LINE MODELING

We compare cross-context generalization performance of the perturbation majority vote, weighted $k$-nearest neighbors, and STATE (Adduri et al., 2025), a representative generative AI baseline. For majority vote and kNN, we train on the three non-target cell lines and test on the fourth, yielding 4 experiments total (one for each possible target cell line). For STATE we also train a separate model for each target cell line. Unlike majority vote and kNN, STATE's design requires training examples from the target cell line, so we train on all four cell lines while holding out only the target's test perturbations. We then generate single-cell predictions for held-out perturbations and call DE using the same procedure (Wilcoxon + BH at FDR $< 0.01$) on predicted vs. control cells. We report accuracy on both the full perturbation–gene pairs and the PerturbQA downsampled subset.

## 2.4 EVALUATION METRICS

We evaluate model performance on accuracy, precision, and F1 score per non-DE, up, and down-regulated class. Per-class accuracy is equivalent to per-class recall in our evaluation since for a class $k$ containing $N_k$ samples, accuracy restricted to those samples equals $\frac{\text{TP}_k}{N_k} = \frac{\text{TP}_k}{\text{TP}_k + \text{FN}_k}$, which is precisely recall for class $k$ (TP = True Positive, FN = False Negative).

## 3 RESULTS

**A simple majority vote achieves high predictive performance on PerturbQA datasets.** The PerturbQA benchmark comprises perturbation-gene pairs filtered for high statistical significance against negative controls (Wu et al., 2025). Despite this curated dataset's widespread use for model development, we find that a simple gene-based majority vote baseline already achieves strong performance. Per-class accuracy, which reduces to per-class recall, remains consistently high across all three classes.(Figure 1a), suggesting that gene-intrinsic response tendency alone is a strong indicator of expression changes in these curated datasets. The performance we gain over SUMMER (LLM method accompanying the PerturbQA data) is largely from higher precision in the `Up` and `Down` classes (see Figure 2).

Applying the identical predictor to the full unfiltered dataset yields drastically different results. Majority vote degenerates to predicting only the dominant non-DE class, achieving near-zero accuracy on up- and down-regulated genes (Figure 1(a); Table 2). This divergence suggests that current perturbation modeling efforts operate on datasets with considerable label noise, complicating the interpretation of reported performance gains and highlighting the need to distinguish between advances in modeling versus the benefits of data curation.

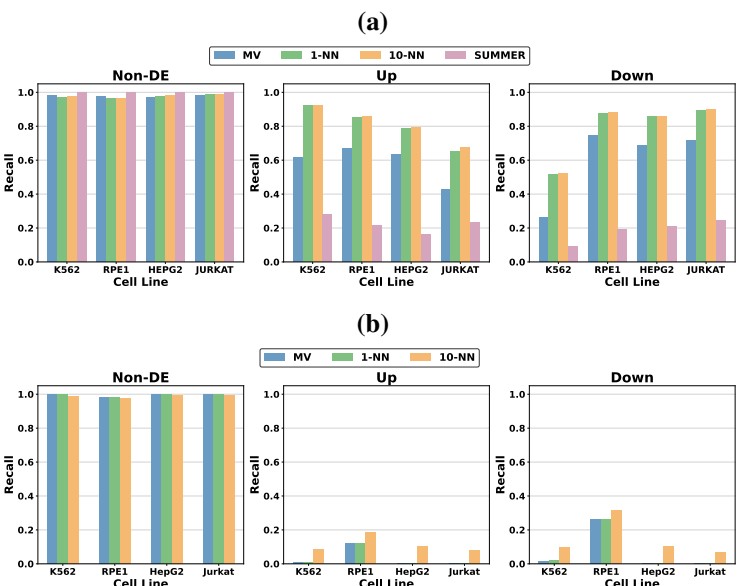

Figure 1: Per-class recall on (a) PerturbQA-downsampled and (b) full datasets.

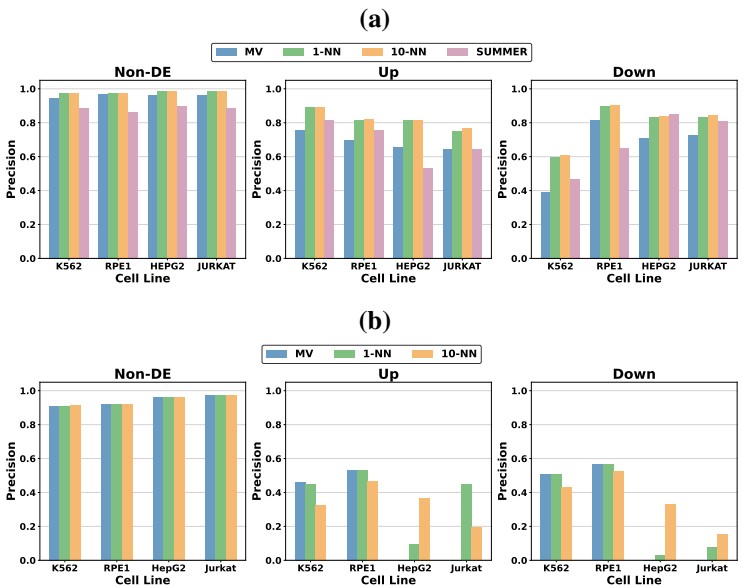

Figure 2: Per-class precision on (a) PerturbQA-downsampled and (b) full datasets.

**Weighted K-nearest neighbors as a simple yet effective model for noisy data.** Given that most perturbation models train on complete datasets, we evaluate whether incorporating perturbation-specific information can recover meaningful performance on unfiltered data. We fit a weighted K-nearest neighbors classifier using GenePT embeddings to represent perturbations – a method requiring no fine-tuning, no large-scale pretraining, and minimal computational resources. Despite operating on the challenging full dataset, 10-NN achieved balanced accuracy of 0.39–0.49 and substantially improved minority class detection: up-regulated F1 increased from near-zero to 0.11–0.27, and down-regulated F1 to 0.09–0.39 (Figure 1(b); Table 2).

To contextualize this performance, we compared against SynthPert (Phillips et al., 2025), which involves fine-tuning LLMs on self-generated chain-of-thought reasoning. On PerturbQA-sampled data, SynthPert reported an average up-regulated F1 of 0.22, which is significantly lower compared to our KNN result on PerturbQA despite orders of magnitude greater complexity (See Figure 1; Figure 2). Our simple distance-weighted voting scheme with a single tuning parameter $k$, cap-

tures much of the signal from perturbation similarity and underscores the importance of establishing strong baselines before pursuing expensive approaches.

### 3.1 CROSS-CELL LINE TRANSFERABILITY

When evaluated on the PerturbQA-downsampled datasets, simple models such as majority vote and kNN outperform complex deep learning models like STATE, which is likely due to STATE requiring larger amount of training data to effectively learn patterns from inputs (Figure 3(a)).

When evaluated on the full datasets without downsampling (Figure 3(b); Table 3), class imbalance exceeds 95% non-DE, causing MV to predict non-DE for nearly all pairs. Both 10-NN and STATE struggle with minority classes, achieving below 6% accuracy on up-regulation and below 18% on down-regulation. STATE shows higher down-regulation accuracy on RPE1 and HepG2, but collapses on Jurkat (up: 0.1%, down: 1.8%), where 10-NN GenePT maintains more stable predictions.

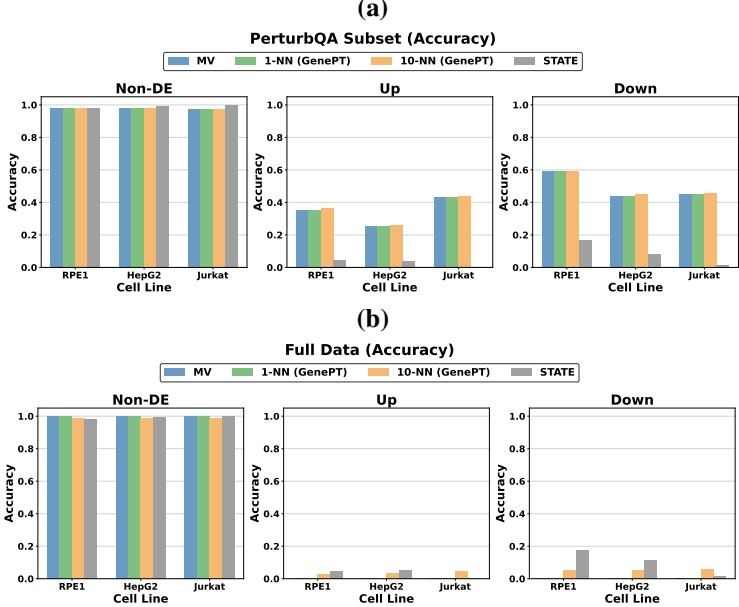

Figure 3: Cross-cell-line transfer per-class accuracy on (a) PerturbQA subset and (b) full data.

## 4 DISCUSSION AND CONCLUSION

Our results show that simple baselines such as embedding-based kNN perform competitively on perturbation outcome prediction. Moreover, the strong performance of majority vote on curated data, paired with its collapse on full data, highlights the importance of careful evaluation. Our baselines do not differentiate confidence levels in the Non-DE/Up/Down labels across training perturbations, particularly for perturbations with few cells. Extending our methods to incorporate uncertainty, handle multi-gene perturbations, and testing on systematic functional-group-based splits would further strengthen our findings. Calibrating continuous models like STATE to produce categorical outputs could also offer the best of both worlds. We advocate for routine inclusion of simple baselines with per-class metrics on Perturb-seq prediction tasks. If the downstream goal is categorical—DE status or cell state changes—modeling these outcomes directly may be more productive than discretizing continuous predictions post hoc.

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

# A APPENDIX

## A.1 ADDITIONAL RESULTS

Table 2: Per-class performance metrics for gene-based majority vote and weighted KNN (k=1, k=10) on full datasets, displayed in Figure 1(b). Nearest neighbor methods use GenePT embeddings to represent perturbations. Best F1 score per dataset for up- and down-regulated classes is bolded.

| Target | Method | Non-DE | | | Up-regulated | | | Down-regulated | | |
|---|---|---|---|---|---|---|---|---|---|---|
| | | Prec. | Recall | F1 | Prec. | Recall | F1 | Prec. | Recall | F1 |
| K562 | MV | 0.909 | 0.999 | 0.952 | 0.463 | 0.007 | 0.014 | 0.506 | 0.017 | 0.034 |
| | 1-NN | 0.909 | 0.999 | 0.952 | 0.450 | 0.007 | 0.015 | 0.507 | 0.019 | 0.036 |
| | 10-NN | 0.915 | 0.985 | 0.949 | 0.327 | 0.086 | **0.137** | 0.432 | 0.098 | **0.160** |
| RPE1 | MV | 0.918 | 0.983 | 0.950 | 0.533 | 0.119 | 0.195 | 0.566 | 0.261 | 0.357 |
| | 1-NN | 0.918 | 0.984 | 0.950 | 0.534 | 0.119 | 0.195 | 0.568 | 0.261 | 0.358 |
| | 10-NN | 0.923 | 0.973 | 0.947 | 0.469 | 0.186 | **0.266** | 0.523 | 0.316 | **0.394** |
| HepG2 | MV | 0.960 | 1.000 | 0.980 | 0.000 | 0.000 | 0.000 | 0.000 | 0.000 | 0.000 |
| | 1-NN | 0.960 | 1.000 | 0.980 | 0.094 | 0.000 | 0.000 | 0.031 | 0.000 | 0.000 |
| | 10-NN | 0.964 | 0.992 | 0.978 | 0.367 | 0.106 | **0.165** | 0.329 | 0.103 | **0.157** |
| Jurkat | MV | 0.974 | 1.000 | 0.987 | 0.000 | 0.000 | 0.000 | 0.000 | 0.000 | 0.000 |
| | 1-NN | 0.974 | 1.000 | 0.987 | 0.448 | 0.003 | 0.005 | 0.077 | 0.001 | 0.003 |
| | 10-NN | 0.976 | 0.991 | 0.983 | 0.193 | 0.078 | **0.111** | 0.155 | 0.068 | **0.094** |

Table 3: Cross-cell-line transfer per-class performance on full datasets. Nearest neighbor methods use GenePT embeddings to represent perturbations. Best F1 score per dataset for up- and down-regulated classes is bolded.

| Target | Method | Non-DE | | | Up-regulated | | | Down-regulated | | |
|---|---|---|---|---|---|---|---|---|---|---|
| | | Prec. | Recall | F1 | Prec. | Recall | F1 | Prec. | Recall | F1 |
| K562 | MV | 0.919 | 1.000 | 0.958 | 0.000 | 0.000 | 0.000 | 0.000 | 0.000 | 0.000 |
| | 1-NN | 0.919 | 1.000 | 0.957 | 0.111 | 0.000 | 0.001 | 0.170 | 0.001 | 0.002 |
| | 10-NN | 0.922 | 0.984 | 0.952 | 0.172 | 0.045 | **0.071** | 0.293 | 0.067 | **0.110** |
| | STATE | | | | | *N/A* | | | | |
| RPE1 | MV | 0.907 | 1.000 | 0.951 | 0.000 | 0.000 | 0.000 | 0.000 | 0.000 | 0.000 |
| | 1-NN | 0.907 | 1.000 | 0.951 | 0.550 | 0.001 | 0.001 | 0.232 | 0.002 | 0.003 |
| | 10-NN | 0.910 | 0.990 | 0.948 | 0.206 | 0.027 | 0.048 | 0.320 | 0.050 | 0.087 |
| | STATE | 0.945 | 0.979 | 0.962 | 0.182 | 0.046 | **0.074** | 0.305 | 0.173 | **0.221** |
| HepG2 | MV | 0.959 | 1.000 | 0.979 | 0.102 | 0.003 | 0.006 | 0.000 | 0.000 | 0.000 |
| | 1-NN | 0.960 | 0.999 | 0.979 | 0.103 | 0.004 | 0.009 | 0.165 | 0.003 | 0.006 |
| | 10-NN | 0.961 | 0.990 | 0.975 | 0.118 | 0.039 | 0.058 | 0.211 | 0.060 | 0.094 |
| | STATE | 0.969 | 0.992 | 0.980 | 0.381 | 0.054 | **0.095** | 0.265 | 0.115 | **0.160** |
| Jurkat | MV | 0.974 | 1.000 | 0.987 | 0.000 | 0.000 | 0.000 | 0.000 | 0.000 | 0.000 |
| | 1-NN | 0.974 | 0.999 | 0.987 | 0.026 | 0.001 | 0.002 | 0.099 | 0.002 | 0.004 |
| | 10-NN | 0.976 | 0.984 | 0.980 | 0.062 | 0.050 | **0.055** | 0.101 | 0.059 | **0.075** |
| | STATE | 0.974 | 0.999 | 0.986 | 0.016 | 0.001 | 0.001 | 0.232 | 0.018 | 0.033 |

