# OpenReview forum: "Rethinking Perturbation Prediction Baselines"
_ICLR.cc/2026/Workshop/FM4Science — ICLR 2026 Workshop FM4Science Poster_

### Official Review · Reviewer_Q97c · 2026-02-14
**Rethinking Perturbation Prediction Baselines - Good paper, accept.**

**Rating:** 7
**Confidence:** 3

**Review:**

The paper investigates the effectiveness of current LLM approaches for predicting cellular responses to genetic perturbations. The authors argue that while recent trends favor complex models, simple baselines for categorical outcomes remain underexplored. They introduce two baselines: a gene-based Majority Vote (MV) and an embedding-based kNN classifier. Evaluating these on Perturb-seq screens across four cell lines, they demonstrate that the MV baseline achieves surprisingly high accuracy on curated benchmarks but collapses on raw, unfiltered data. Crucially, they show that a 10-NN classifier using GenePT embeddings remains competitive with state-of-the-art generative models like STATE, especially in cross-cell-line transfer tasks, despite having orders of magnitude fewer parameters.

Strengths

S1: Critical Evaluation of Benchmarks: The paper provides a reality check on existing perturbation benchmarks. By showing how MV collapses on unfiltered data, the authors expose how dataset curation can artificially inflate model performance.

S2: Principled Baseline Development: The introduction of MV and kNN as baselines is well-justified. These baselines help distinguish between a model's ability to capture meaningful perturbation-specific signals versus simply memorizing gene-intrinsic properties.

S3: Efficiency and Practicality: The 10-NN baseline with GenePT embeddings achieves competitive results without the need for extensive fine-tuning or massive computational resources, making it a highly practical alternative for researchers.

S4: Biological Interpretability: By shifting focus from continuous gene expression values to categorical DE outcomes, the work prioritizes results that are more directly interpretable for biologists.

Weaknesses

W1: Scope of Categorical Outcomes: While categorical prediction (Up/Down/Non-DE) is biologically useful, the paper could benefit from discussing how much information is lost by ignoring the magnitude of expression changes, which is often critical for understanding dosage-response curves.

W2: Limited Comparison to Traditional Methods: While comparing against LLM-based methods and generative models is relevant, the paper lacks a comparison to traditional differential expression analysis tools or simpler linear models often used in transcriptomics.

W3: Generalization to Diverse Perturbations: The study focuses on essential gene screens. It remains unclear if these simple baselines would hold up as well for more complex chemical perturbations or combinatorial genetic knockdowns where synergistic effects are less likely to be captured by a gene-intrinsic majority vote.

W4: Limited Tasks: The evaluation is restricted to categorical classification. The absence of continuous metrics (e.g., Pearson/Spearman correlation of expression vectors) makes it difficult to assess if these baselines capture the biological nuance required for drug discovery.

W5: Limited Data Diversity: The datasets used are all CRISPRi screens in immortalized lines; the performance of the kNN baseline on high-variance single-cell data or more heterogeneous tissue samples remains unproven.

---

### Official Review · Reviewer_VcSL · 2026-02-14
**A crucial reality check for perturbation prediction: strong empirical evidence with constructive suggestions on embedding audits.**

**Rating:** 7
**Confidence:** 4

**Review:**

Summary: This paper performs a critical re-evaluation of "perturbation prediction" methods in Perturb-seq, focusing on categorical differential expression outcomes. The authors demonstrate that simple baselines (gene-wise majority vote and embedding-based kNN) perform comparably to or better than complex state-of-the-art models on standard curated benchmarks.

Strengths:

1. The paper provides a much-needed critical perspective on the field. Demonstrating that complex deep learning models do not significantly outperform simple kNN baselines is a significant scientific discovery that challenges current trends and prevents wasted community effort.

1. The distinction between performance on curated vs. unfiltered data is a key contribution. It convincingly argues that future method development must account for data selection biases to be genuinely useful.

1.  The proposed baselines are simple, interpretable, and easy to implement. They serve as excellent lower-bound metrics for any future research claiming to advance the state-of-the-art in perturbation prediction.

Weaknesses:

1.  Since the kNN baseline relies heavily on embeddings, the validity of the conclusion depends entirely on the quality and independence of these embeddings. The paper needs to be explicitly clear about how these embeddings were generated. If they utilize a Foundation Model pre-trained on similar data, the kNN performance might be inflated due to implicit leakage. A detailed audit or ablation using "naive" embeddings (e.g., PCA on raw counts) would strengthen the claim.

1. While the simple baselines are effective, there is a gap between the trivial "Majority Vote" and the non-parametric "kNN". Adding a simple parametric linear baseline (e.g., Logistic Regression or a linear probe on the embeddings) would be beneficial. If a linear model also performs well, it would strongly suggest that the underlying biological problem is less complex than assumed, further reinforcing the paper's central thesis.

1.  The observation that performance collapses on unfiltered data is important, but currently lacks actionable guidance. Rather than a deep causal analysis (which might be out of scope), the paper would benefit from a correlation analysis: which specific filtering criteria (e.g., read depth, mitochondrial content) correlate most with the performance drop? This would help the community define "minimally viable" curation standards.

---

### Official Review · Reviewer_hdXh · 2026-02-18
**Simple Baselines Expose Evaluation Pitfalls in Genetic Perturbation Prediction**

**Rating:** 8
**Confidence:** 3

**Review:**

This paper challenges recent trends in genetic perturbation prediction by demonstrating that simple baselines, such as majority voting and embedding-based k-nearest neighbors, achieve competitive performance against complex deep learning and LLM-based models, particularly when evaluated on realistic, unfiltered datasets.

**Pros**
- The paper provides a critical reality check for the field by revealing that high performance on popular curated benchmarks (like PerturbQA) can be achieved by a majority vote baseline that ignores perturbation identity entirely.
- The distinction made between "downsampled" (curated) and "full" (realistic) evaluation settings effectively highlights how severe class imbalance in biological data is often overlooked in current model evaluations.
- The proposed weighted k-nearest neighbors classifier using GenePT embeddings offers a computationally efficient alternative that outperforms complex generative models like STATE in detecting minority classes on full datasets.
- The study convincingly argues for the necessity of including simple, interpretable baselines in future research to prevent the overestimation of algorithmic progress.

**Cons**
- The scope is limited to categorical prediction of differential expression status, neglecting the prediction of continuous expression magnitudes which is essential for understanding the potency of perturbations.
- While the k-nearest neighbors approach is effective, the paper provides limited analysis on the sensitivity of the method to different embedding types beyond GenePT or varying values of $k$.
- The comparison focuses primarily on STATE and specific LLM-based methods, and could be strengthened by benchmarking against a broader range of recent graph-based or transformer-based perturbation models.

---

### Meta-Review · Area_Chair_bkzg · 2026-02-28

**Recommendation:** Accept (Oral)
**Confidence:** 4

**Metareview:**

This paper re-evaluates perturbation prediction in Perturb-seq datasets and demonstrates that simple baselines—gene-wise majority vote and embedding-based kNN—match or outperform recent LLM-based and generative approaches under realistic, unfiltered evaluation settings. Reviewers highlight the importance of distinguishing curated versus full datasets and appreciate the paper’s role as a methodological reality check for the field. The results suggest that current benchmarks may overstate progress and that stronger baselines are needed before scaling model complexity.

While the scope is limited to categorical differential expression outcomes and additional baseline comparisons would strengthen the conclusions, the empirical evidence is compelling and the contribution is scientifically valuable.

---

### Decision · Program_Chairs · 2026-03-03

Accept (Poster)